# Modulation of Hair Growth Promoting Effect by Natural Products

**DOI:** 10.3390/pharmaceutics13122163

**Published:** 2021-12-15

**Authors:** Seyeon Park, Joomin Lee

**Affiliations:** 1Department of Applied Chemistry, Dongduk Women’s University, Seoul 02748, Korea; sypark21@dongduk.ac.kr; 2Department of Food and Nutrition, Chosun University, Gwangju 61452, Korea

**Keywords:** alopecia, natural product, hair growth

## Abstract

A large number of people suffer from alopecia or hair loss worldwide. Drug-based therapies using minoxidil and finasteride for the treatment of alopecia are available, but they have shown various side effects in patients. Thus, the use of new therapeutic approaches using bioactive products to reduce the risk of anti-hair-loss medications has been emphasized. Natural products have been used since ancient times and have been proven safe, with few side effects. Several studies have demonstrated the use of plants and their extracts to promote hair growth. Moreover, commercial products based on these natural ingredients have been developed for the treatment of alopecia. Several clinical, animal, and cell-based studies have been conducted to determine the anti-alopecia effects of plant-derived biochemicals. This review is a collective study of phytochemicals with anti-alopecia effects, focusing mainly on the mechanisms underlying their hair-growth-promoting effects.

## 1. Introduction

Hair affects the impacts of thermoregulation, physical protection, and social interaction [1]. Hair disorders, which include alopecia, anagen effluvium, telogen effluvium, hirsutism, hypertrichosis, and miniaturization, negatively affect health [2]. Alopecia is defined as a decrease in hair density and can be considered a common symptom of several pathologies, such as inflammation and infection [3]. Alopecia can be a sign of a systemic disease, including thyroid dysfunction, systemic lupus erythematosus, trichotillomania, or infection [3]. Although alopecia is a non-life-threatening condition, it can affect the esthetics, social activities, and quality of life of individuals [4]. Hair growth is the result of the growth and differentiation of hair follicles (HFs) comprising dermal papilla cells (DPCs) and epithelial cells. The formation of HF involves four cycles: anagen (growth), catagen (regression), telogen (rest), and exogen (shedding). The hair cycle transition is controlled by several growth stimulatory or inhibitory factors [5]. Alopecia can be categorized into several classes. Androgenetic alopecia, the most common form of alopecia, is a male or female pattern hair loss that is characterized by a progressive loss of hair diameter, length, and pigmentation. It is caused by genetic factors and inadequate androgen signaling [6]. Androgenetic alopecia also worsens conditions such as drug side effects, acute stressors, and weight loss [6]. Alopecia areata is an autoimmune disorder (similar disorders include thyroid disease, celiac disease, vitiligo, and atopy) caused by lymphocytes that attack the bulb of HFs in the anagen phase [7]. This disease affects up to 2% of the population and commonly occurs during childhood or adolescence [7]. Patients with alopecia areata may experience sudden hair loss [8]. Telogen effluvium refers to the loss of telogen hair due to abnormal hair circulation, and generally 100–200 telogen hairs are lost every day [9]. The causes of acute telogen effluvium (in which hair loss lasts less than six months) includes systemic disease, drugs, fever, psychological/emotional stress, weight loss, childbirth, iron and vitamin D deficiency, inflammatory scalp disorders, interruption of oral contraceptives, and iron deficiency [10]. Primary cicatricial alopecia, known as scarring alopecia, encompasses hair loss disorders in which the hair follicle is irreversibly destroyed [11]. Secondary cicatricial alopecia occurs from irreversible loss of hair follicles due to thermal burns, metastatic cancer, trauma, or radiation [11]. To date, the US Food and Drug Administration (FDA)-approved non-surgical treatment alternatives for hair loss include drugs such as minoxidil and finasteride. Minoxidil is converted to minoxidil sulfate by sulfotransferase present in the scalp, which promotes the growth of hair follicular cells and reduces hair loss [12]. Finasteride is a 5α-reductase inhibitor that blocks the conversion of testosterone to dihydrotestosterone (DHT), which causes androgenetic alopecia [13]. However, these drugs have side effects in patients. Thus, it is necessary to identify new, safe, and effective drugs to treat hair loss.

Recently, interest in hair loss prevention using natural products or their extracts has increased. Products currently marketed as using natural ingredients include those used to prevent hair loss in the form of hair tonics, hair growth promoters, hair conditioners, and hair cleansers [14]. Plants and their extracts contain multiple components, such as polyphenols, flavonoids, terpenoids, carotenoids, and fatty acids, which support the maintenance of HF health [15]. Plant-based formulations have the advantage of being easy to acquire from low-cost materials, and their non-toxic effects have been noted since ancient times. Although natural products are widely used to prevent hair loss, little is known about their exact mechanisms of action. The present study is a review of the molecular mechanisms underlying the hair-promoting effects of various herbs and their constituents.

## 2. Biochemical Action of Herbs and Their Extracts

Various phytochemicals and their active constituents have been shown to promote hair growth in vivo and in vitro. Table 1 provides an overview of bioactive components from plants. A summary of the potential mechanisms of action on hair growth using plants is presented in Figure 1 and Table 2.

### 2.1. Growth Factors

Numerous growth factors were expressed in the HFs. Fibroblast growth factor (FGF), vascular endothelial growth factor (VEGF), keratinocyte growth factor (KGF), insulin-like growth factor (IGF), epidermal growth factor (EGF), and hepatocyte growth factor (HGF) act as hair growth stimulators, and transforming growth factor-β1 (TGF-β1) is a hair growth suppressor [6]. VEGF is secreted by DPCs and is involved in hair growth through the formation of new blood vessels around follicles [58,59]. VEGF expression is decreased in HFs during alopecia compared to that in normal follicles [59]. IGF-1 signaling has been reported to influence HF development and tissue renewal [60]. IGF-1 prevents HFs from entering the catagen phase [61,62]. The FGF family consists of 22 members and regulates a variety of biological functions [63]. Basic FGF (FGF-2) and KGF-2 (FGF-10) stimulate HF growth [64]. KGF mediates growth, development, and differentiation of HF [6]. HGF is involved in the stimulation of mouse follicle growth and HF elongation in vitro and in vivo [65,66]. TGF-β1 and its receptors are involved in the catagen phase of the hair cycle and promote tissue remodeling and apoptosis [67].

Sinapic acid exhibits various biological activities, including antioxidant, anti-inflammatory, anticancer, anti-hyperglycemic, and neuroprotective effects [68,69,70,71,72,73]. Sinapic acid treatment has been reported to increase VEGF and IGF-1 expression and increase the proliferation of human HF-derived dermal papilla cells (hHFDPCs) [16]. Icariin is a bioactive compound from Epimedium brevicornum Maxim extract, which possesses testosterone mimetic properties and anti-osteoporotic and antidepressant-like effects [74,75,76]. Treatment with icariin increased IGF-1 secretion in vibrissae follicles and upregulated IGF-1 mRNA and protein levels in DPCs [17]. Geranium sibiricum L., belonging to the family Geraniaceae, has been used globally as an antioxidant and anti-inflammatory substance [77,78,79]. Geranium sibiricum extract increased HGF and VEGF expression, and decreased TGF-β1 expression in vitro and in vivo [18]. Oleuropein, an olive constituent, is a phenolic glycoside that possesses several pharmacological properties, including antioxidant, antimicrobial, anticancer, cardioprotective, and neuroprotective effects [80,81,82]. Topical administration of oleuropein (0.4 mg/day) significantly upregulated IGF-1, KGF, HGF, and VEGF mRNA expression in mouse skin tissue compared with control mice [19]. Caffeine is a well-known stimulant that is widely consumed in common beverages [83]. It acts as a phosphodiesterase inhibitor and possesses biological activities, including antioxidant and cancer preventive effects [83,84,85,86]. Treatment with caffeine (0.001%) and testosterone (5 μg/mL) significantly increased IGF-1 expression but decreased TGF-β2 expression in human outer sheath keratinocytes [20]. *Carthamus tinctorius* L., known as Safflower, has been used in various medical conditions [87,88]. *Carthamus tinctorius* L. extract (CTE) decreased 5α-reductase activity and promoted hair growth in mice [89]. CTE increased VEGF and KGF mRNA expression and decreased TGF-β1 expression in vitro [21]. Red ginseng oil (RGO) extracted from red ginseng possesses antioxidant [90] and anti-inflammatory [91,92] effects, and its major constituents include linoleic acid (LA), β-sitosterol (SITOS), and bicyclo(10.1.0)tridec-1-ene (BICYCLO) [93]. Truong et al., revealed that RGO (10%) exhibited hair regeneration capacity in a testosterone-induced androgenic alopecia C57BL/6 mouse model [22]. RGO and its main compounds decreased the expression of TGF-β1 compared with testosterone treatment [22]. Quercitrin (quercetin-3-O-rhamnoside) is a flavonoid found in various plants and has been shown to protect against cisplatin-induced hair damage [94]. Quercitrin treatment resulted in an increase in bFGF, KGF, platelet-derived growth factor (PDGF)-AA, and VEGF mRNA and protein levels in hDPCs [23]. *Sophora flavescens* possesses various pharmacological properties, including anti-inflammatory, anti-arthritic, and antioxidant effects [95,96]. It has been demonstrated that *Sophora flavescens* extract promoted hair growth by inducing mRNA expression of IGF-1 and KGF in cultured DP cells [24]. Shikimic acid is commercially used in cosmetics and has been shown to possess antibacterial, anti-inflammatory, antifungal, anti-aging, and whitening effects [97]. A recent study revealed that shikimic acid upregulated the mRNA expression of HGF, KGF, and VEGF in hDPCs [25]. Procyanidin (PC)-B3 is a procyanidin dimer that has been studied for its hair-growth-promoting effect [98,99]. The study showed increased hair-growing activity in vitro and anagen-inducing activity in vivo, as well as a potential inhibitory effect of TGF-β1 [26]. Ginsenosides present in Ginseng Radix et Rhizoma are Rb1, Rb2, Rb3, Rd, Re, Rg1, Rg3, and Rh2, and ginsenoside Rb1 is one of the active compounds present in ginseng [100]. Studies have reported the hair-promoting effects of ginseng extract and ginsenosides in vitro and in vivo [100]. Ginsenoside Rb1 treatment induced VEGF-A and VEGF receptor 2 and attenuated TGF-β1 expression [27]. Nelumbinis Semen (NS) is a widely used functional food that contains nutritional compounds with therapeutic benefits [101,102]. NS improved oxidative stress on the scalp of hair loss patients due to its high content of total polyphenols and flavonoids [103]. Park et al. also showed that NS extract possessed a strong antioxidant capacity and may reduce the oxidative damage that causes hair loss [28]. NS-extract-treated mice showed increased VEGF and IGF-1 mRNA expression [28]. However, TGF-β1 mRNA expression was decreased after NS extract treatment compared to that after dimethyl sulfoxide (DMSO) treatment [28]. *Chamaecyparis obtusa* (CO), belonging to the family Cupressaceae, has alpha-terpinyl acetate, sabinene, isobornyl acetate, and limonene as major constituents [104]. An essential oil from CO has shown anti-inflammatory and antimicrobial activities in previous studies [104,105,106,107]. In the CO-treated mice group, IGF-1 mRNA expression was increased compared with that in the group treated with 3% minoxidil for 4 weeks. VEGF expression was also upregulated in the skin of mice after CO treatment [29]. Polygonum multiflorum (PM) has been used in Chinese medical practices [108] and reported to have various actions, including antioxidant [109], anti-human-immunodeficiency-virus (HIV) [110], neuroprotective [111,112], and hepatoprotective effects [113]. 2,3,5,4-Tetrahydroxystilbene-2-O-D-glucoside (TSG), a major component of PM, induces new hair growth in C57BL/6J mice [113]. Additionally, Shin et al., observed that PM extract (20 μg/mL) increased IGFBP2, PDGF, and VEGF expression in cultured hDPCs [30]. Alnus sibirica Fisch. ex Turcz (AS), belonging to the family Betulaceae, is rich in flavonoids [114], tannins [115,116], and diarylheptanoids [117]. AS or oregonin (active substance in AS) treatment increased IGF-1 levels and decreased TGF-β1 levels in H_2_O_2_-induced stressed hDPCs [31]. *Malva verticillate* (MV) is an edible plant widely used in East Asia [118]. MV is a rich source of phenolic compounds that possess antioxidants [119], anti-complementary, hypoglycemic [120], and antidiabetic effects by activating AMP-activated protein kinase [121]. MV seed extract treatment upregulated the mRNA expression of growth factors, including IGF-1, KGF, VEGF, and HGF [32]. In another study, LA in MV seeds also elevated IGF-1, KGF, VEGF, and HGF mRNA expression in cultured hDPCs [33]. Liposomal honokiol is a natural extract from *Magnolia officinalis* [122,123] and mainly possesses anticancer effects [124,125]. Li et al. demonstrated that it has a hair-promoting effect, including increased thickness of the dermis and the number of HF in C57BL/6N mice [34]. Liposomal honokiol treatment inhibited TGF-β1 protein expression and phosphorylated SMAD2 expression in the outer root sheath (ORS), as determined using immunohistochemistry analysis [34].

### 2.2. Cytokines

Cytokines such as interleukin (IL)-1α, IL-1β, tumor necrosis factor-α (TNF-α), interferon-γ (IFN-γ), IL-2, IL-4, and IL-5 can influence the hair cycle [126]. IL-1α, IL-1β, and TNF-α are potent inducers of hair loss [127,128]. These cytokines exhibit similar patterns in alopecia areata, which involves abnormal keratinization of the hair matrix [129]. Overexpression of IL-lα in transgenic mice led to the development of inflammatory skin diseases, such as hair loss [130]. C57BL/6 mice overexpressing TNF-α, IL-1β, and IFN-γ promote keratinocyte apoptosis associated with hair loss [131]. Clinical data revealed elevated serum levels of IL-4 in patients with localized alopecia areata [132].

*Angelica gigas* Nakai (AGN) has been extensively studied as a medicinal plant [133]. In particular, the roots of AGN showed antinociceptive activity in pain models [134], and neuroprotective [135] and beneficial effects in treating ischemia [136]. A recent study demonstrated that treatment with decursin (0.15%), a major component isolated from AGN root, or AGN root extract (2%) for 17 days stimulated hair growth in vivo [35]. These treatments reduced the protein levels of pro-inflammatory cytokines (TNF-α and IL-1β) and increased anti-inflammatory cytokines (IL-4 and IL-13) in the dorsal skin of mice [35]. 3-Deoxysappanchalcone (DSC) is a biologically active compound from Caesalpinia sappan L., which has been suggested to have anti-inflammatory, anticancer, and anti-allergic effects [137,138,139]. 3-DSC treatment (0.1–3 μM) increased IL-6-mediated signal transducer and activator of transcription (STAT) 3 expression in hDPCs [36]. 3-DSC also inhibits the phosphorylation of STAT6 mediated by IL-4 [36]. Another study reported that subjects with androgenic alopecia treated with shampoo containing Inula helenium and Caesalpinia Sappan extract (3-DSC) showed increased hair density and hair count [140]. Broussonetia papyrifera (BP), also known as paper mulberry, is a medicinal herb that utilizes leaves, fruits, and bark [141]. The polysaccharide extract from the fruits of BP showed antioxidant and antibacterial activities [142]. The BP root extract contains flavonoids, which results in inhibitory effects on nitric oxide, inducible nitric oxide synthase, TNF-α, and IL-6 [143]. Treatment with BP eliminated IL-4-induced STAT6 phosphorylation in hDPCs [37].

### 2.3. Wnt/β-Catenin

Wnt/β-catenin signaling is broadly utilized and plays a crucial role in HF morphogenesis [144]. In canonical Wnt/β-catenin signaling, Wnt proteins bind to Frizzled (FZD) receptors and low-density lipoprotein receptor-related protein 5/6 (LRP5/6) co-receptors. Once activated, the β-catenin complex with APC and Axin is phosphorylated by casein kinase Iα (CKIα) and glycogen synthase kinase-3β (GSK-3β). Therefore, β-catenin accumulates in the cytoplasm and binds to the transcription factor T-cell factor/lymphoid enhancing factor (TCF/LEF) in the nucleus [145]. The study reported that Wnt/β-catenin activation promotes hair regeneration by inhibiting GSK-3β in hDPCs [146]. Sinapic acid treatment increased the protein level of β-catenin by upregulating phosphorylated GSK-3β and Akt [16]. Oleuropein elevated nuclear β-catenin protein expression and increased LEF1 and cyclin D1 mRNA expression in DPCs [19]. Oleuropein administration increased the mRNA levels of Wnt10B, LRP5, and FZDR1, and the protein level of β-catenin in mice compared to minoxidil-treated C57BL/6 mice [19]. RGO remarkably increased the protein expression of β-catenin and LEF1 induced by testosterone and RGO co-treatment in dorsal skin tissues compared to treatment with testosterone alone [22]. RGO also enhanced β-catenin and phosphorylated GSK-3β protein expression in C57BL/6 mouse skin compared to that in the minoxidil-treated group [38]. Thuja orientalis (TO) is used to treat dermatitis, gout, and chronic tracheitis [147]. TO (5.05 mg/cm^2^/day) administration promoted hair growth and the early anagen phase, and prolonged the mature anagen phase in mice [39]. Immunohistochemical analysis showed increased β-catenin expression after TO treatment [39]. 3,4,5-Tri-O-caffeoylquinic acid (TCQA) is a caffeoylquinic acid derivative that acts as a neuroprotective agent and protects against amyloid-β (Aβ)-induced cell death [148,149]. Topical treatment with 1% TCQA prolonged anagen phase induction in C3H mice for 30 days [40]. It was also confirmed that β-catenin expression increased in the skin of TCQA-treated mice and HFDPCs [40]. Gene expression profile data revealed an increase in the expression of the canonical Wnt-associated genes, Ctnnb1, Wls, Wnt2b, and Wnt4 after TCQA treatment [40]. *Ishige sinicola* (IS) is a brown alga that exhibits various activities, including osteoblastic bone formation [150] and anti-inflammatory effects [151]. IS treatment for 3 weeks increased hair-fiber length in rat vibrissa follicles and induced anagen progression of the hair shaft [41]. Protein expression of β-catenin and phosphorylation of GSK3β were increased after treatment with IS in cultured DPCs [41]. Prunus mira Koehne (PK) is a wild peach species that contains various nutrients and fatty acids [152]. Zhou et al., showed the hair-promoting effect of nut oil from PK [42]. They demonstrated that nut oil from PK increased hair length in mice and upregulated Wnt10B, β-catenin, and GSK-3β expression in mice [42]. Costunolide, a constituent of *Saussurea lappa*, has been used as an antioxidant, anti-inflammatory, and anticancer agent [153,154,155]. Costunolide increased cell proliferation and β-catenin expression in hHFDPCs [43]. Morroniside is the main component of *Cornus officinalis* and possesses neuroprotective, anti-apoptotic, and antioxidant effects [156,157,158,159]. Morroniside increased Wnt/β-catenin signaling by upregulating Wnt10B, β-catenin, and LEF1 in cultured human ORS cells [44]. In a mouse model, morroniside promoted the anagen phase and delayed the catagen phase of HF, which was partly related to an increase in β-catenin expression [44]. 3-DSC inhibited the phosphorylation of β-catenin protein but promoted the transcriptional activity of TCF/LEF [36]. Timosaponin BII extracted from Anemarrhena asphodeloides rhizome is known to have antioxidant [160], anti-inflammatory [161,162], and anticancer [163] properties. A clinical study showed that application of timosaponin BII (0.5%) containing scalp care solution for 28 days resulted in improvement in hair and scalp conditions, particularly hair luster, scalp hydration, hair fall number, and scalp redness level [164]. A recent study showed that timosaponin BII (0.5%) increased the hair regrowth area and HF number in mice [45]. Moreover, timosaponin BII treatment was shown to upregulate β-catenin and Wnt10B expression in the dorsal skin of mice [45]. The application of 20 mg/mL liposomal honokiol also increased Wnt3a and β-catenin expression [34]. *Malva verticillata* (MV) is a traditional herb native to Mongolia that contains flavan-3-ols, flavonoids, and fatty acids [119]. MV is a therapeutic candidate for diabetes [121], bone disease [165], and cancer [166]. MV leaves, stems, and seeds have been shown to be a rich source of phenolic compounds. Microbial fermentation has been used to increase the extraction yield of bioactive compounds from natural products [167]. Bacillus subtilis fermentation of MV leaves exhibited an antioxidant and osteogenic effect [168]. The seeds of MV (50 μg/mL) activated β-catenin protein expression in cultured DPCs [32]. Myristoleic acid, an active compound of MV, upregulates Wnt reporter activity [32]. Another study showed that LA in MV seeds increased cell proliferation and phosphorylation of GSK-3 and β-catenin in DPCs [33]. Salvia plebeia (SP) belongs to the family Labiaceae and is used for its antioxidant [169], anti-inflammatory [170,171], and anti-influenza [172] effects. It is reported to contain flavonoids, phenolic acids, and other nutrients [169]. SP extract enhanced the proliferation of hHFDPC and increased the TCF/LEF-luciferase activity as well as the level of β-catenin protein expression [46]. The use of hair tonics, including Broussonetia papyrifera (BP) extract, showed an increase in total hair count for 12 weeks in a clinical study. In addition, BP treatment increased TCF/LEF-luciferase activity and β-catenin protein levels in vitro [37]. *Undariopsis peterseniana* (UP), an edible brown seaweed, is a rich source of nutrients and acts as an antioxidant and anti-inflammatory agent [173,174,175,176]. UP extract was shown to induce hair growth in ex vivo organ cultures [47]. UP extract upregulated the phosphorylation levels of β-catenin and GSK-3β compared with the control in DPCs [47].

### 2.4. 5α-Reductase Inhibitory Effect

Testosterone is converted to DHT by 5α-reductase, which is known to cause androgenic alopecia [177]. Androgenic alopecia affects over 50% of men over the age of 50 [178], compared to only 25% of women by the age of 49 and 41% of women by the age of 69 years [179]. Finasteride, a 5α-reductase inhibitor, has shown improvement in androgenic alopecia in clinical trials [13]. However, the use of 5-α reductase inhibitors has revealed that it has sexual and reproductive side effects [180]. DHT binds to androgen receptors in DPCs, leading to the onset of the telogen phase [177]. *Sophora flavescens* or *Undariopsis peterseniana* treatment showed potent 5α-reductase inhibitory effects [24,47]. Octaphlorethol A, a constituent of Ishige sinicola, inhibited 5α-reductase activity compared to that of finasteride [41]. Costunolide treatment downregulated testosterone-induced 5α-reductase mRNA expression in hDPCs [43]. Puerariae Flos (PF) has been used as a medicinal herb for its antioxidant, antidiabetic, and protective effects against ethanol-induced injury [181,182,183]. PF extract showed inhibitory effects on 5α-reductase and hair-growth-promoting effects in mice [48]. Cacumen platycladi (CP) is a Chinese medicine containing organic acids, flavonoids, and phenylpropanoids [184]. Treatment with CP decreased DHT levels and 5α-reductase expression while promoting hair growth in vivo [49]. Another study showed that CP volatile oil treatment increased the proliferation of hDPCs and shortened the time of hair regrowth [185]. Ginseng rhizomes have been used in medical remedies [186,187]. Ginsenoside Ro, a major ginsenoside constituent in the ginseng rhizome, has anti-inflammatory [188,189], antioxidant [190], and anti-obesity [191] effects. Murata et al., revealed that red ginseng rhizome extract, ginsenoside Ro, and ginsenoside Rg3 showed inhibitory effects on 5α-reductase activity [50]. The study also showed that topical administration of red ginseng rhizomes (2 mg/mouse) and ginsenoside Ro (0.2 mg/mouse) induced hair regrowth in testosterone-treated mice [50]. Physcion, a component of Polygonum multiforum (PM), has anti-inflammatory, antioxidant, and anticancer effects [192,193]. Treatment with physcion exhibited hair growth-promoting activity in testosterone*-*treated C57BL/6 mice [51]. In addition, physcion inhibits 5α-reductase expression in vivo [51]. Rosmarinus officinalis belongs to the Lamiaceae family and has been widely studied for its antibacterial, antioxidant, and anticancer activities [194,195,196]. Murata et al. showed that Rosmarinus officinalis leaf extract improved hair regrowth in mice, as well as 5α-reductase inhibitory activity [52]. *Avicennia marina* is used in traditional medicine for the treatment of skin diseases, rheumatism, ulcers, and smallpox [197]. Avicequinone C, isolated from *Avicennia marina*, attenuated 5α-reductase inhibitory activity (IC_50_ of 38.8 ± 1.29 µM) [53]. Ecklonia cava, an edible marine brown alga, contains a variety of bioactive compounds, including phlorotannins, carotenoids, and fucoidans [198]. *Ecklonia cava* has been reported to have various biological properties, including antioxidant, anti-inflammatory, anti-allergy, and anticancer effects [199]. Shin et al., showed that *Ecklonia cava* polyphenols reduced oxidative stress in hDPCs [54]. It was found that androgens, the main cause of androgen alopecia, increase reactive oxidative species (ROS) in hDPCs in which androgen receptors are overexpressed [54]. In addition, TGF-β-1 secretion induced by androgen was inhibited by an ROS scavenger, indicating that antioxidants can promote hair growth [54]. Topical application of 0.5% *Ecklonia cava* enzymatic extract induced anagen progression on the back of C57BL/6 mice [200]. Furthermore, the *Ecklonia cava* enzymatic extract, dieckol, inhibited 5α-reductase activity [200].

### 2.5. Sonic Hedgehog (Shh) Signaling

The hedgehog pathway is one of the most important signaling pathways in tissue development, homeostasis, and repair [201]. Hedgehog signaling is triggered by Sonic hedgehog (Shh), Indian hedgehog (Ihh), and Desert hedgehog (Dhh). Shh signaling plays an important role in HF development [202]. Hedgehog signaling occurs through the binding of hedgehog ligands to the receptor Patched (PTCH) 1, which suppresses the activation of Smoothened (SMO). Inhibition of SMO results in the translocation of the glioma-associated (GLI) gene to the nucleus [202]. Shh plays an essential role in the cell–cell interactions involved in the morphogenesis of hair follicles [203]. Hair follicle development results from complex signaling between epithelial and mesenchymal cells. Various signaling pathways are involved in hair follicle development, such as Wnt, bone morphogenetic protein (BMP), platelet-derived growth factor (PDGF), Notch, and ectodysplasin, and they exhibit signaling crosstalk with the hedgehog pathway [201]. RGO treatment upregulated Shh signaling-related expression of Shh, SMO, and GLI1 in testosterone-induced C57BL/6 mice [22] and in mouse skin [38]. TO treatment resulted in an increase in Shh expression [39]. Costunolide elevated GLI1 mRNA and protein expression in hDPCs [43]. Epigallocatechin-3-gallate (EGCG) is a major bioactive molecule in green tea that has been shown to act on multiple molecular targets to ameliorate various human diseases [204]. Green tea leaf extracts have polyphenolic components that exhibit anti-inflammatory and stress-inhibitory effects, which may influence mouse hair growth [205]. EGCG treatment promoted the growth of mink hair follicles and the proliferation of DPCs and outer root sheath cells (ORSCs) [55]. EGCG has also been shown to increase the protein levels of Shh, PTCH, Smo, and Gli1 in hair follicles [55].

### 2.6. Apoptosis

Apoptosis plays an important role in morphological development and is accompanied by a number of characteristic morphological changes, including cell shrinkage, nuclear condensation, and cellular fragmentation [206]. There are two major apoptosis signaling pathways: the death receptor (extrinsic) pathway and the mitochondria-mediated pathway [207]. The extrinsic pathway is initiated by cell-surface-expressed death receptors of the tumor necrosis factor superfamily. Once the receptor is activated, caspase-8 is activated and initiates apoptosis by direct cleavage of downstream effector caspases [207]. The intrinsic pathway is initiated by intracellular stresses, and it induces permeabilization of the outer mitochondrial membrane and activates the mitochondrial pathway. Once mitochondrial permeabilization occurs, cytochrome c is released into the cytosol and associates with caspase-9 and Apaf-1 to form apoptosomes, which can activate caspase-3 or caspase-7, causing apoptosis [207]. Apoptosis signaling is attenuated by a particular group of proteins (Bcl-2, Bcl-xL, and Mcl-1) [208]. Another group of proteins (Bax, Bak, Bok, Bim, Bad, Bcl-xS, and Bid) act as apoptotic agonists that promote apoptosis [208]. In the anagen phase, hair follicular cells undergo dynamic cell proliferation and differentiation to form the hair shaft [209]. During the catagen phase, growth factors expressed by DPCs cause a decrease in the proliferation and differentiation of hair matrix keratinocytes, leading to apoptosis [210,211]. The anagen-catagen transition plays a clinically important role in human hair growth; the abnormal termination of the anagen phase has been shown to cause gradual hair thinning [211]. Therefore, extension of the anagen phase is a key strategy for the prevention of hair loss. Caffeine administration inhibits apoptosis and necrosis in human ORS keratinocytes [20]. The decrease in the protein level of Bcl-2 in testosterone-treated mice was reversed after treatment with RGO, LA, and SITOS [22]. Treatment with RGO after UVC exposure inhibited cleaved caspase-3, cleaved caspase-9, and cleaved poly-ADP ribose polymerase (PARP) expression in mouse skin tissues [38]. In addition, RGO treatment resulted in an increased Bax/Bcl-2 ratio in the UVC-treated group compared to that in the control group [38]. Quercitrin administration resulted in a decrease in mRNA expression of Bad, although the mRNA and protein expression of Bcl-2 increased after treatment of cultured hDPCs with 10 nM and 100 nM quercitrin [23]. PM extract also showed an increase in Bcl-2 mRNA expression and a decrease in Bad expression in hDPCs [30]. Panax ginseng extract inhibited apoptosis in Dickkopf-1 (DKK-1)-induced ORS keratinocytes as determined by terminal deoxynucleotidyl transferase-mediated dUTP nick end labeling (TUNEL) staining [56]. Moreover, Panax ginseng extract downregulated the mRNA expression of Bcl-2 and upregulated the mRNA expression of Bax in DKK-1-induced ORS keratinocytes and HF [56]. AS treatment increased protein expression of Bcl-2 but led to a decrease in the protein levels of Bax, PARP, and caspase-3 in HFDPCs [31]. SP extract elevated the protein expression of Bcl-2 and decreased Bax expression compared with the negative or positive control (10  μM minoxidil) in hDPCs [46]. The extract from the berries Serenoa repens (SR), commonly known as saw palmetto, showed therapeutic effects as a 5α-reductase inhibitor [212]. SR and DHT co-treatment stimulated hair growth compared to that induced by DHT in vivo [57]. Moreover, SR treatment led to a decrease in the protein expression of TGF-β2, cleaved caspase-3, and Bax, but increased Bcl-2 expression compared to DHT treatment [57].

### 2.7. Cell Cycle

Cell proliferation is controlled by factors that regulate the transition between different cell cycle stages in mammalian cells [213]. Cell cycle progression also plays a major role in HF biology [214]. The cell cycle consists of four phases: gap phase 1 (G1), DNA synthesis (S), gap phase 2 (G2), and mitosis (M) [214]. Cyclins and cyclin-dependent kinases (CDKs) function as regulators of the G1/S or G2/M phases [215]. Cyclin D1 binds to CDK4 and CDK6 and drives cell cycle progression into the G1 phase [213]. Cyclin E associates with CDK2 to promote G1-S phase transition [214]. Sinapic acid treatment was accompanied by an increase in cyclin D1 and the distribution of cells in the G0/G1 phase, as well as a decreased distribution in the S and G2/M phases [16]. Cyclin D1 is a direct target for transactivation by the β-catenin/LEF-1 pathway through an LEF-1 binding site in the cyclin D1 promoter and is a direct downstream molecule in the β-catenin pathway [213]. Woo et al. suggested that sinapic acid treatment increased cell growth and cell cycle progression through an increase in cyclin D1 expression [16]. Administration of RGO increased the protein expression of cyclin D1 and cyclin E in testosterone-treated mice [22] and C57BL/6 mouse models [38]. UP treatment upregulated cyclin D1, phospho(ser780)-pRB, cyclin E, phospho*-*CDK2, and CDK2 protein expression in DPCs [47]. Treatment with IS upregulated cyclin E and CDK2 expression in cultured DPCs [41]. LA in MV seeds increased the mRNA levels of cyclin D1 and CDK2 in vitro [33]. EGCG treatment increased the number of cells in S phase, and upregulated the protein levels of cyclin D1 and cyclin B1 [55].

## 3. Conclusions

This study reviewed the beneficial effects of herbs and their bioactive compounds on hair growth, and their underlying mechanisms of action (growth factors and cytokines, Wnt/β-catenin, 5α-reductase inhibitory effect, sonic hedgehog signaling, apoptosis, and cell cycle progression). The herbs and their constituents investigated in this study act via multiple signaling mechanisms in the prevention of alopecia. Therefore, they have the potential to be more effective than minoxidil and finasteride, which are conventionally used to treat hair loss. In this review, we have attempted to provide a database of phytochemicals for hair-growth-promoting effects. This information will serve as a basis for developing more effective therapeutic agents for the treatment of alopecia and improving our understanding of their mechanisms of action.

## Figures and Tables

**Figure 1 pharmaceutics-13-02163-f001:**
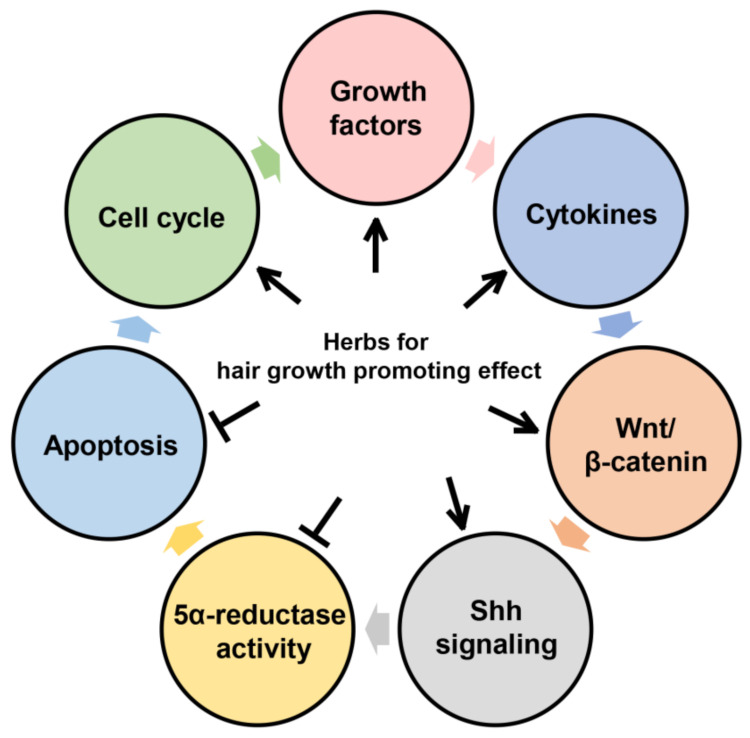
Interactions between herbs and their mechanisms for hair-growth-promoting efficacies.

**Table 1 pharmaceutics-13-02163-t001:** Bioactive components from plants with hair-growth-promoting therapeutic effects.

Botanical Name	Parts Used	Bioactive Components	Type of Extract	Ref
*Citrus limon* L., *Fragaria ananassa* L., *Secale cereale* L.	-	Sinapic acid	-	[16]
*Epimedium* spp.	Leaves	Icariin	-	[17]
*Geranium sibiricum* L.	Whole plant	Corilagin, gallic acid	Methanolic extract	[18]
*Olea europaea* L.	Unprocessed olive fruit and leaves	Oleuropein	-	[19]
*Camellia sinensis*, *Coffea ara* *bica*	-	Caffeine	-	[20]
*Carthamus tinctorius* L.	Floret	Hydroxysafflor yellow A	Ethanolic extract	[21]
*Panax ginseng* Mayer	Whole plant	Linoleic acid, β-sitosterol, bicyclo(10.1.0)tridec-1-ene	Supercritical fluid extraction system	[22]
*Hottuynia cordata* Thunb.	-	Quercitrin	-	[23]
*Sophora flavescens* Aiton	Roots	L-maackiain, medicarpin	Methanolic extract	[24]
*Illicium anisatum* L., *Illicium verum* Hook. f.	-	Shikimic acid	*-*	[25]
*Hordeum vulgare* L. var. *distichon* Alefeld	Seed	Procyanidin B-3	Aceton extract	[26]
*Panax ginseng* Mayer	-	Ginsenoside Rb1	-	[27]
*Nelumbinis semen*	Whole plant	Anthraquinone, flavonoids, tannin, saponins	Ethanolic extract	[28]
*Chamaecyparis obtusa*	-	α-Terpinyl acetate, sabinene, isobornyl acetate, limonene	Oil	[29]
*Polygonum multiflorum*	Roots	2,3,5,4′-Tetrahydroxystilbene2-O-β-D-glucoside, emodin	Ethanolic extract	[30]
*Alnus sibirica* Fisch. ex Turcz	Whole plant	Oregonin	Ethanolic extract	[31]
*Malva verticillate*	Seed	Myristoleic acid	Ethanolic extract	[32]
*Malva verticillate*	Seed	Linoleic acid	Ethanolic extract	[33]
*Magnolia officinalis*	-	Liposomal honokiol	-	[34]
*Angelica giga*s Nakai	Roots	Decursin	Ethanolic extract	[35]
*Caesalpinia sappan* L.	-	3-Deoxysappanchalcone	-	[36]
*Broussonetia papyrifera*	Whole plant	7-hydroxycoumarin, protocatechuate acid, ferulic acid, protocatechuic acid and epicatechin	Ethanolic extract	[37]
*Panax ginseng* Mayer	Whole plant	Linoleic acid, β-sitosterol	Supercritical fluid extraction system	[38]
*Thuja orientalis*	Leaves	Kaempferol, isoquercetin	Hot water extract	[39]
*Ipomoea batatas* L.	-	3,4,5-tri-O-caffeoylquinic acid	-	[40]
*Ishige sinicola*	Whole plant	Octaphlorethol A	Ethanolic extract	[41]
*Prunus mira* Koehne	Nut oil	α-tocopherol, vitamin E β-sitosterol, linoleic acid, oleic acid	Pressing the seeds	[42]
*Saussurea lappa* Clarke	-	Costunolide	-	[43]
*Cornus officinalis*	-	Morroniside	-	[44]
*Anemarrhena asphodeloides*	-	Timosaponin BII	-	[45]
*Salvia plebeia* R. Br.	Whole plant	Flavonoids, monoterpenoids, sesquiterpenoids, diterpenoids, triterpenes, phenolic acids	Methanolic extract	[46]
*Undariopsis peterseniana*	Whole plant	Apo-9′-fucoxanthinone	Ethanolic extract	[47]
*Pueraria thomsonii*	Whole plant	Soyasaponin I, kaikasaponin III	Ethanolic extract	[48]
*Platycladus orientalis* (L.) Franco	Leaves	Myricitrin, isoquercitrin, quercitrin, myricetin, afzelin, quercetin, kaempferol, amentoflavone, hinokiflavone	Ethanolic extract, aqueous extract	[49]
*Panax ginseng*	Rhizome	Ginsenoside Ro	Ethanolic extract	[50]
*Polygonum multiforum* Thunb.	Leaves	Physcion	-	[51]
*Rosmarinus officinalis*	Leaves	12-methoxycarnosic acid	Ethanolic extract	[52]
*Avicennia marina*	Heartwood	Avicequinone C	Methanolic extract	[53]
*Ecklonia cava*	Whole plant	Dieckol	Enzymatic hydrolysis reaction	[54]
*Camellia sinensis* L. Ktze.	-	Epigallocatechin-3-Gallate	-	[55]
*Panax ginseng* Mayer	Roots	Ginsenosides	Ethanolic extract	[56]
*Sabal serrulatum*	-	Fatty acids, phytosterols	-	[57]

**Table 2 pharmaceutics-13-02163-t002:** Studies of hair-growth-promoting effects using natural products or their extracts.

Natural Products	Experimental Model	Treatment	Effects	Ref
Sinapic acid	Human hair-follicle-derived papilla cells	10, 50, and 100 μM	Induction of cell proliferation and cell cycle progression; activation of Akt and GSK-3β/β-catenin signaling; increased expression of VEGF and IGF-1	[16]
Icariin	Cultured vibrissae follicles	10 and 20 μM	Induction of hair shaft elongation and prolonged anagen phase; increase of IGF-1 production and expression	[17]
	Cultured dermal papilla cells	10 and 20 μM		
	C57BL/6 mice	0.01		
*Geranium sibiricum* extract	Human dermal papilla cells	9.8–156.3 ppm	Induction of cell proliferation and migration; induced expression of Ki-67 protein, HGF, and VEGF in vitro; reduced number of mast cells and the expression of TGF-β1 in mouse skin	[18]
	C57BL/6 mice	1000 ppm		
Oleuropein	Human follicle dermal papilla cells	10, 20, and 50 μM	Induction of cell proliferation; increase of LEF1 and cyclin D1 mRNA expression and β-catenin protein expression in vitro; induction of anagenic hair growth and Wnt/β-catenin pathway in vivo; upregulation of IGF-1, KGF, HGF, and VEGF gene expression in mice	[19]
	C57BL/6 mice	0.4 mg/mouse/day		
Caffeine	Cultured hair follicles Human hair-follicle-derived outer root sheath keratinocytes (ORSKs)	0.0005% 0.00001, 0.0001, 0.001%	Increase of hair shaft elongation, anagen duration; increase of hair matrix keratinocyte proliferation and IGF-1 expression in hair follicles; increase of cell proliferation and IGF-1 expression in RSKs; inhibited apoptosis/necrosis and TGF-β2 protein secretion in RSKs	[20]
*Carthamus* *Tinctorius* Floret extract	Human keratinocytes (HaCaT)	0.005–1.250 mg/mL	Induction of cell proliferation in dermal papilla cells and HaCaT increase of VEGF, KGF; de crease of TGF-β1; increase of length of cultured hair follicles and stimulated the growth of hair in mice	[21]
	Human hair follicle-derived papilla cells	0.005-1.250 mg/mL		
	Cultured hair follicles	50, 100 and 200 μg/mL		
Red Ginseng Oil	C57BL/6 mice	10%	Increase of hair growth; upregulated β-catenin, Lef-1, Sonic hedgehog, Smoothened, Gli-1, Cyclin D1, and Cyclin E expression; reduced the protein level of TGF-β; enhanced the expression of Bcl-2	[22]

Quercitrin	Human dermal papilla cells Cultured hair follicles	0.1, 1, 10, 100 nM and 1 μM 5 and 10 μM	Enhanced the cell viability and cellular energy metabolism; increase of expression of Bcl-2 and Ki67; upregulation of bFGF, KGF, PDGF-AA and VEGF; stimulated hair shaft growth in cultured hair follicles	[23]

*Sophora* *flavescens* extract	Cultured dermal papilla cells	10^−6^, 10^−5^ and 10^−4^%	Induction of hair growth in vivo; increase of IGF-1 and KGF in vitro; decrease of 5a-reductase activity in vivo	[24]
	Sprague-Dawley rats	0.001, 0.01 and 0.01%		
	C57BL/6 mice	1%		
Shikimic acid	Human follicle dermal papilla cells Human outer root sheath keratinocytes C57BL/6 mice Cultured hair follicles	0.1, 1, 10, 100 μM and 1 mM 1 and 10 μM 10 and 100 mM 1 and 10 μM	Induction of hair growth in vivo; increase of Cell proliferation in hDPCs and hORSCs; en hanced hair shaft elongation in cultured hair follicles; increased c-myc, HGF, KGF, VEGF, p38 MAPK and CREB	[25]
Procyanidin B-3	Cultured hair epithelial cells from C3H/HeNCrj mice	0.1–100 μg/mL	Increased hair-growing activity in vitro and anagen-inducing activity in vivo; potential inhibitory effect of TGF-β1	[26]
	C3H mice	200 μL/day/mouse		
Ginsenoside Rb1	Cultured mink hair follicles	5 and 10 μg/mL	Increase of the growth of hair follicles; upregulated the expression levels of VEGF-A and VEGF-R2, while attenuated the TGF-β1 expression; activation of PI3K/AKT/GSK-3β signaling pathway in hair follicles and DPCs.	[27]
	Cultured dermal papilla cells	10 μg/mL		
*Nelumbinis Semen* extract	Human follicle dermal papilla cells	15.63–125 ppm	Enhanced cell proliferation and migration; high mRNA expression of VEGF and IGF-1; low TGF- β1 mRNA expression	[28]
	C57BL/6 mice	1000 ppm		
*Chamaecyparis**obtusa* oil	C57BL/6 mice	3%	Increase of ALP and γ-GT activities in the skin tissue; increase of IGF-1 mRNA expression; increase of VEGF and decrease of EGF expression in the skin tissue; increase of SCF expression	[29]

*Polygonum* *multiflorum* extract	Human follicle dermal papilla cells Cultured hair follicles	10 and 100 μg/mL 2, 20, and 50 μg/mL	Increased cell viability and mitochondrial activity; increase of Bcl-2 and decrease of BAD and DKK-1; increase of IGFBP2, PDGF and VEGF; prolonged the anagen of human hair follicles	[30]
*Alnus* *sibirica* Fisch. ex Turcz	Human follicle dermal papilla cells	22, 66 and 200 μg/mL	Inhibition of apoptosis; increased IGF-1 and decreased TGF-β1 expression; decreased DHT production	[31]
*Malva verticillata* seed extracts	Human follicle dermal papilla cells	10–100 μg/mL	Increased Wnt reporter activity; increased β-catenin level; increased IGF-1, KGF, VEGF and HGF	[32]
Linoleic acid in *Malva verticillate* seed	Human follicle dermal papilla cells	3, 10 and 30 μg/mL	Activated Wnt/β-catenin signaling; induced cell growth by increasing the expression of cyclin D1 and CDK2; increased VEGF, IGF-1, HGF and KGF; inhibited DKK-1	[33]
Liposomal honokiol	C57BL/6 mice	20 mg/mL	Promoted hair regrowth; accelerated the hair growth cycle by up regulating the Wnt3a/β- catenin signaling pathway; inhibited the TGF-β1/p-smad2 signaling pathway during the anagen stage	[34]

Decursin, *Angelica* *Giga*s Nakai root extract	C57BL/6 mice	2%	Induction of hair growth; decrease of TNF-and IL-1β; increase of IL-4 and IL-13	[35]

3-Deoxysappanchalcone	Human follicle dermal papilla cells C57BL/6 mice	0.1–10 µM	Increased cell proliferation; increase of β-catenin and Tcf; increase of IL-6-induced phosphorylation and subsequent transactivation of STAT3, thereby increasing the expression of Cdk4, FGF and VEGF; promoted the anagen phase of hair growth in C57BL/6 mice	[36]
		3 mM		
*Broussonetia* *papyrifera*	Human follicle dermal papilla cells NIH3T3 cells Clinical Study	1.25–40 μg/mL 1–40 μg/mL Korean males and females	Promoted cell proliferation; enhanced TCF/LEF-luciferase activity and increased the level of β-catenin protein; inhibited IL4-induced STAT6 phosphorylation; increased hair count after using the hair tonic for 12 weeks	[37]
Red Ginseng Oil	C57BL/6 mice SKH-1 hairless mice	50% 1%	Induction of hair growth; upregulated expression of β-catenin, phospho-GSK3β, Lef-1, Gli-1, Smoothened, Cyclin D1, Cyclin E, IGF-1 and VEGF; protective effect against UVC-induced skin damage in SKH-1 hairless mice by inhibiting inflammation and apoptosis	[38]
*Thuja orientalis*	C57BL/6 mice	5.05 mg/cm^2^/day	Induction of hair growth including hair number and size of hair follicles; induction of β-catenin and Shh protein expression	[39]
3,4,5-tri-O-caffeoylquinic acid	Human follicle dermal papilla cells C3H mice	5, 10, 15 and 25 μM 1%	Increase of β-catenin in vitro and in vivo; upregulation in hair growth-associated genes using microarray	[40]
*Ishige sinicola*	Cultured rat vibrissa follicles C57BL/6 mice Sprague-Dawley rats	1, 10, and 100 μg/mL 0.1, 1, and 10 μg/mL 0.1, 1, and 10 μg/mL	Induction of anagen progression of the hair shaft; inhibition of 5α-reductase activity; in crease of cell proliferation; increase of phos pho-GSK3β, β-catenin, Cyclin E, and CDK2, and decrease of p27kip1	[41]
*Prunus* *mira* Koehne	C57BL/6 mice	15.06, 30.13, and 60.26 mg/cm^2^/day	Increase of hair growth; increase of Wnt 10b, β-catenin, and GSK-3β	[42]
	KM mice	0.47–60.26 mg/cm^2^/day		
Costunolide	Human follicle dermal papilla cells C57BL/6 mice	0.1, 0.3, 1, and 3 μM 0.3 μM	Increase of cell proliferation in vitro; inhibited the 5a-reductase activity in hHFDPCs; in creased the level of β-catenin and Gli1 and de creased TGF-β1; increase of hair length in vivo	[43]
Morroniside	Cultured outer root sheath cells C57BL/6 mice	1 and 10 µM 100 µM	Enhanced outer root sheath cell proliferation and migration in vitro; upregulation of Wnt10b, β-catenin, and LEF1; accelerated the onset of anagen and delayed hair follicle cat agen	[44]
Timosaponin BII	C57BL/6 mice	0.5%	Induced hair growth; increased expression of β-catenin and Wnt10b	[45]
*Salvia* *plebeian* extract	Human follicle dermal papilla cells C57BL/6 mice	7.8, 15.6, and 31.3 μg/mL 1,000 μg/mL	Increased cell proliferation; increased HGF and decreased TGF-β1 and SMAD2/3; inhibi tion of apoptosis by increasing the Bcl-2/Bax ratio; enhanced hair growth in mice	[46]
*Undariopsis* *peterseniana*	C57BL/6 mice Cultured rat vibrissa follicles Sprague-Dawley rats NIH3T3 fibroblasts Dermal papilla cells	0.1, 1, and 10 μg/mL 1, 10, and 100 μg/mL 0.1, 1, 10, and 100 μg/mL 0.1, 1, 10, and 100 μg/mL 0.1, 1, and 10 μg/mL	Increase of the hair-fiber lengths and anagen initiation in vivo; decreased 5α-reductase ac tivity and increased cell proliferation in vitro; increased the levels of Cyclin D1, phos pho(ser780)-pRB, Cyclin E, phospho-CDK2, and CDK2; increase of phosphorylation of ERK and the levels of Wnt/β-catenin signaling proteins	[47]
Puerariae Flos	Wistar/ST rats C57BL/6NCrSlc mice C3H/He mice	50, 200, and 500 μg/mL 2 and 5 mg/mouse/day 2 and 5 mg/mouse/day	Increase of hair re-growth effect in testos terone-treated C57BL/6NCrSlc and C3H/He mice; inhibitory activity of against testos terone 5a-reductase	[48]
*Cacumen* *platycladi*	Sprague-Dawley rats C57BL/6NCrSlc(C57) mice	0.02–2.5 μmol/L 2 and 5 mg/mouse/day	Increase of hair growth; increase of Wnt 10b, β-catenin, and GSK-3β	[49]
Ginseng rhizome, Ginsenoside Ro	Wistar rats C57BL/6 mice	Ginseng rhizome (200, 500, and 1,000 μg/mL) Ginseng rhizomes (2 mg/mouse), and ginsenoside Ro (0.2 mg/mouse)	Induced hair re-growth in vivo; inhibitory effects against 5αR	[50]
Physcion	Sprague-Dawley ratsC57BL/6 mice	12.5–100 µM5 mg/mouse/day	Inhibited the 5a-reductase activity; hair -growth-promoting activity	[51]
*Rosmarinus* *officinalis* leaf extract	Wistar rats C3H/He mice C57BL/6 mice LNCaP cells	50, 200, and 500 μg/mL 2 mg/mouse/day 2 mg/mouse/day 1–5 µM	Improved hair regrowth in C57BL/6NCrSlc mice; increased hair growth in C3H/He mice that had their dorsal areas shaved; decreased 5a-reductase activity; inhibited androgen-de pendent proliferation of LNCaP cells	[52]
Avicequinone C	Human hair dermal papilla cells	5 and 10 µg/mL	Decreased 5a-reductase activity	[53]
*Ecklonia cava* enzymatic extract, dieckol	Cultured rat vibrissa follicles Sprague-Dawley rats C57BL/6 miceRat vibrissa, immortalized dermal papilla cell, NIH3T3 cells	0.01–10 µg/mL 0.5% 0.5% 0.001–10 µg/mL 0.05 and 10 µg/mL	Increased hair-fiber length and anagen pro gression of the hair-shaf after *E. cava* enzy matic extract; decreased 5a-reductase activity in the presence of dieckol	[54]
Epigallocatechin-3-Gallate	Cultured mink hair follicles Cultured dermal papilla cells Cultured outer root sheath cells	0.1–5 µM 0.25–4 µM 0.25–4 µM	Promoted hair follicle growth in DPCs and ORSCs; activated Shh and Akt signaling; in creased the expression of cyclin D1 and cyclin E1	[55]
*Panax* *ginseng* extract	Cultured outer root sheath keratinocytes	20 ppm 20 ppm	Increased proliferation and inhibited apopto sis in ORS keratinocytes; abrogated DKK-1-in duced growth inhibition of cultured HFs ex vivo	[56]
*Serenoa repens* extracts	Human keratinocyte cells C57BL/6 mice	1, 5, 25, and 100 μg/ mL 50%	Increased cell viabilities; stimulated hair folli cle growth; decreased inflammatory response; decreased TGF-β2 and cleaved caspase-3 ex pression of hair loss mouse; inhibited apopto sis	[57]

## Data Availability

Not applicable.

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
