# Peer review of "Modulation of Hair Growth Promoting Effect by Natural Products"

_pharmaceutics, 2021, doi:10.3390/pharmaceutics13122163_

Round 1

Reviewer 1 Report

This review describe modulation of hair growth by natural compounds. 

This review have to be improved in lenght!! 

Pls expand introduction, it is too much short. 

Also section relative to drugs/natural compounds is only touch on, need to strongly improve it (in particular 2.4-2.7)

Lack scheme or figures that could help readers to understand concepts. Mandatory add some... 

Hence, this reviewer indicate accept this MS for publication after major revisions

Author Response

December 01, 2021

Editor-in-Chief

pharmaceutics

Manuscript ID pharmaceutics-1476843 entitled " Modulation of hair growth promoting effect by natural products”

Dear Editor,

Thank you for your willingness to consider a revision of the above referenced manuscript. We appreciate the careful review and constructive suggestions. We are now submitting the revised version of our manuscript, which has been prepared according to the reviewers’ suggestions. Our responses follow, addressing specific reviewers’ comments, and providing descriptions of the changes made during revision.

Response to Reviewer 1 Comments

Point 1: This review have to be improved in lenght!! Also section relative to drugs/natural compounds is only touch on, need to strongly improve it (in particular 2.4-2.7)

Response 1: Thank you for your comment, and we completely agree with the reviewer. As suggested, we improved the length in revised manuscript. (Please check the part marked in red in the manuscript)

Point 2: Lack scheme or figures that could help readers to understand concepts. Mandatory add some...

Response 2: As suggested, we added the table 1 including botanical name, plant sources, bioactive components and type of extract in revised manuscript.

Again, we appreciate the opportunity to revise our work for consideration for publication in pharmaceutics. We hope our revision meets your approval. Thank you for taking the time and effort to help us improve the paper.

Sincerely,

Joomin Lee, Ph.D.

Reviewer 2 Report

Journal: Pharmaceutics

Review: "Modulation of hair growth promoting effect by natural products"

  1. In Table 1 in the first column, the column title is inappropriate with the content. In the column,  there are pure compounds names, extracts names and  plants names. This must be corrected as uniform form. According to the aim of this work, the title of the first  column should be rewritten, for example: “Plant Sources" or "botanical names of plants"; here it is necessary to add the plant parts that used. Another column should be added as "Bioactive compounds or extracts"; to give the main important bioactive compounds known in the plants or their bioactive extracts. OR make two table one for plants with their botanical names and other table for constituents.
  2. I suggest to write the title "Dose" instead of the column title "treatment".
  3. What are the traditional formulations that  prepared from the used plants. Or add the type of extracts in the table.
  4. Give more details about diseases decrease the hair growth.
  5. There are many other important references studied the hair growth related to medicinal plants, but they are not mentioned or discussed in this manuscript.
  6. Line 48, 64: "in vivo and in vitro"; check to write in italic form across the manuscript.
  7. Line 67: "Sinapic acid treatment…". To the compounds mentioned in the manuscript, should be added the names of their botanical sources.

Author Response

December 01, 2021

Editor-in-Chief

pharmaceutics

Manuscript ID pharmaceutics-1476843 entitled " Modulation of hair growth promoting effect by natural products”

Dear Editor,

Thank you for your willingness to consider a revision of the above referenced manuscript. We appreciate the careful review and constructive suggestions. We are now submitting the revised version of our manuscript, which has been prepared according to the reviewers’ suggestions. Our responses follow, addressing specific reviewers’ comments, and providing descriptions of the changes made during revision.

Response to Reviewer 2 Comments

Point 1: In Table 1 in the first column, the column title is inappropriate with the content. In the column, there are pure compounds names, extracts names and plants names. This must be corrected as uniform form. According to the aim of this work, the title of the first column should be rewritten, for example: “Plant Sources" or "botanical names of plants"; here it is necessary to add the plant parts that used. Another column should be added as "Bioactive compounds or extracts"; to give the main important bioactive compounds known in the plants or their bioactive extracts. OR make two table one for plants with their botanical names and other table for constituents.

Response 1: We thank the reviewer for raising an important issue that escaped our attention. As suggested, we added the table 1 including botanical name, plant sources, bioactive components and type of extract in revised manuscript.

Point 2: I suggest to write the title "Dose" instead of the column title "treatment".

Response 2: As suggested, we changed “dose” instead of "treatment" in Table 2.

Point 3: What are the traditional formulations that  prepared from the used plants. Or add the type of extracts in the table.

Response 3: As suggested, As suggested, we added the table 1 including botanical name, plant sources, bioactive components and type of extract in revised manuscript.

Point 4: Give more details about diseases decrease the hair growth.

Response 4: As suggested, we mention the hair loss diseases in revised manuscript. (Please see the line 26-50)

Point 5: There are many other important references studied the hair growth related to medicinal plants, but they are not mentioned or discussed in this manuscript.

Response 5: As suggested, we added the more important references in revised manuscript.

Point 6: Line 48, 64: "in vivo and in vitro"; check to write in italic form across the manuscript.

Response 6: As suggested, we changed the word in italic form. (Please see the line 26-50)

Point 7: Line 67: "Sinapic acid treatment…". To the compounds mentioned in the manuscript, should be added the names of their botanical sources.

Response 7: As suggested, the biological sources for all plants mentioned in our manuscript are listed in Table 1.

Again, we appreciate the opportunity to revise our work for consideration for publication in pharmaceutics. We hope our revision meets your approval. Thank you for taking the time and effort to help us improve the paper.

Sincerely,

Joomin Lee, Ph.D.

Reviewer 3 Report

Dear Authors,

After the review process, I have several comments: the introduction is too poor in the details; you should include new findings based on the correlations between bioactivity and bioavailability of functional compounds; you should make an association with other physiological processes where phenolic compounds (from plant and/or mushrooms extracts) could have positive effects; also, you should base their comments and paper ideas on the bioactive potential of functional products and bioavailability of phenolic compounds.

Best regards.

Author Response

December 01, 2021

Editor-in-Chief

pharmaceutics

Manuscript ID pharmaceutics-1476843 entitled " Modulation of hair growth promoting effect by natural products”

Dear Editor,

Thank you for your willingness to consider a revision of the above referenced manuscript. We appreciate the careful review and constructive suggestions. We are now submitting the revised version of our manuscript, which has been prepared according to the reviewers’ suggestions. Our responses follow, addressing specific reviewers’ comments, and providing descriptions of the changes made during revision.

Response to Reviewer 3 Comments

Point 1: After the review process, I have several comments: the introduction is too poor in the details; you should include new findings based on the correlations between bioactivity and bioavailability of functional compounds.

Response 1: As suggested, As suggested, we improved the length in revised manuscript.

Point 2: you should make an association with other physiological processes where phenolic compounds (from plant and/or mushrooms extracts) could have positive effects; also, you should base their comments and paper ideas on the bioactive potential of functional products and bioavailability of phenolic compounds.

Response 1: Thank you for your comment, and we completely agree with the reviewer. We added the biological sources for all plants mentioned in our manuscript. (Please see Table 1)

Again, we appreciate the opportunity to revise our work for consideration for publication in pharmaceutics. We hope our revision meets your approval. Thank you for taking the time and effort to help us improve the paper.

Sincerely,

Joomin Lee, Ph.D.

Round 2

Reviewer 1 Report

As suggested Authors have improved paper and increase length of some parts

Remain to add AT LEAST one graphical scheme to explain much better some concept

Hence this reviewer indicate accept this MS for publication after minor revisions

Author Response

December 9, 2021

Editor-in-Chief

pharmaceutics

Manuscript ID pharmaceutics-1476843 entitled " Modulation of hair growth promoting effect by natural products”

Dear Editor,

Thank you for your willingness to consider a revision of the above referenced manuscript. We appreciate the careful review and constructive suggestions. We are now submitting the revised version of our manuscript, which has been prepared according to the reviewers’ suggestions. Our responses follow, addressing specific reviewers’ comments, and providing descriptions of the changes made during revision.

Response to Reviewer 1 Comments

Point 1: As suggested Authors have improved paper and increase length of some parts. Remain to add AT LEAST one graphical scheme to explain much better some concept

Response 1: Thank you for your comment, and we completely agree with the reviewer. As suggested, we added graphical scheme in revised manuscript. (Please check figure 1)

Again, we appreciate the opportunity to revise our work for consideration for publication in pharmaceutics. We hope our revision meets your approval. Thank you for taking the time and effort to help us improve the paper.

Sincerely,

Joomin Lee, Ph.D.

Reviewer 2 Report

pharmaceutics

Review article: "Manuscript ID pharmaceutics-1476843 entitled " Modulation of hair growth promoting effect by natural products”

  1. In the last review we suggested to check across the manuscript to write the terms in vivo and in vitro into italic form. However, many of them still not corrected across the manuscript; such in Lines: 86, 99, …………etc.
  2. In table 2; the title "plant source" is not correct or not suitable for the content of the first column, which contains compounds or extracts. This can be called as (natural products).

Author Response

December 09, 2021

Editor-in-Chief

pharmaceutics

Manuscript ID pharmaceutics-1476843 entitled " Modulation of hair growth promoting effect by natural products”

Dear Editor,

Thank you for your willingness to consider a revision of the above referenced manuscript. We appreciate the careful review and constructive suggestions. We are now submitting the revised version of our manuscript, which has been prepared according to the reviewers’ suggestions. Our responses follow, addressing specific reviewers’ comments, and providing descriptions of the changes made during revision.

Response to Reviewer 2 Comments

Point 1: In the last review we suggested to check across the manuscript to write the terms in vivo and in vitro into italic form. However, many of them still not corrected across the manuscript; such in Lines: 86, 99, …………etc.

Response 1: We thank the reviewer for raising an important issue that escaped our attention. As suggested, we changed the word “in vivo” and “in vitro” into italic form in revised manuscript.

Point 2: In table 2; the title "plant source" is not correct or not suitable for the content of the first column, which contains compounds or extracts. This can be called as (natural products).

Response 2: As suggested, we changed “natural products” instead of " plant source" in Table 2.

Again, we appreciate the opportunity to revise our work for consideration for publication in pharmaceutics. We hope our revision meets your approval. Thank you for taking the time and effort to help us improve the paper.

Sincerely,

Joomin Lee, Ph.D.

Reviewer 3 Report

Dear Authors,

The relation between functional products and bioavailability was not so well mentioned. I recommend reading my first review comments and making the necessary corrections. Best regards.

Author Response

December 09, 2021

Editor-in-Chief

pharmaceutics

Manuscript ID pharmaceutics-1476843 entitled " Modulation of hair growth promoting effect by natural products”

Dear Editor,

Thank you for your willingness to consider a revision of the above referenced manuscript. We appreciate the careful review and constructive suggestions. We are now submitting the revised version of our manuscript, which has been prepared according to the reviewers’ suggestions. Our responses follow, addressing specific reviewers’ comments, and providing descriptions of the changes made during revision.

Response to Reviewer 3 Comments

Point 1: The relation between functional products and bioavailability was not so well mentioned. I recommend reading my first review comments and making the necessary corrections.

Response 1: We agree with the reviewer and have mentioned  in the revised manuscript (please see line 142-145, 260-263, 314-322).

Again, we appreciate the opportunity to revise our work for consideration for publication in pharmaceutics. We hope our revision meets your approval. Thank you for taking the time and effort to help us improve the paper.

Sincerely,

Joomin Lee, Ph.D.
